# High-Temperature Corrosion Behavior of Selected HVOF-Sprayed Super-Alloy Based Coatings in Aggressive Environment at 800 °C

**DOI:** 10.3390/ma16124492

**Published:** 2023-06-20

**Authors:** Zdeněk Česánek, Kateřina Lencová, Jan Schubert, Jakub Antoš, Radek Mušálek, František Lukáč, Marek Palán, Marek Vostřák, Šárka Houdková

**Affiliations:** 1Research and Testing Institute Plzen, 30100 Pilsen, Czech Republic; cesanek@vzuplzen.cz (Z.Č.); lencova@vzuplzen.cz (K.L.); schubert@vzuplzen.cz (J.S.); antos@vzuplzen.cz (J.A.); houdkova@vzuplzen.cz (Š.H.); 2Institute of Plasma Physics of CAS, 18200 Prague, Czech Republic; musalek@ipp.cas.cz (R.M.); lukac@ipp.cas.cz (F.L.); 3Department of Materials and Engineering Metallurgy, University of West Bohemia, 30100 Pilsen, Czech Republic; marek@palan.cz

**Keywords:** hot corrosion, molten salt, HVOF, Ni-based coatings, Co-based, gravimetric method, Na_2_SO_4_/NaCl, MCrAlYX

## Abstract

This study is focused on the high-temperature corrosion evaluation of selected thermally sprayed coatings. NiCoCrAlYHfSi, NiCoCrAlY, NiCoCrAlTaReY, and CoCrAlYTaCSi coatings were sprayed on the base material 1.4923. This material is used as a cost-efficient construction material for components of power equipment. All evaluated coatings were sprayed using HP/HVOF (High-Pressure/High-Velocity Oxygen Fuel) technology. High-temperature corrosion testing was performed in a molten salt environment typical for coal-fired boilers. All coatings were exposed to the environment of 75% Na_2_SO_4_ and 25% NaCl at the temperature of 800 °C under cyclic conditions. Each cycle consisted of 1 h heating in a silicon carbide tube furnace followed by 20 min of cooling. The weight change measurement was performed after each cycle to establish the corrosion kinetics. Optical microscopy (OM), scanning electron microscopy (SEM), and elemental analysis (EDS) were used to analyze the corrosion mechanism. The CoCrAlYTaCSi coating showed the best corrosion resistance of all the evaluated coatings, followed by NiCoCrAlTaReY and NiCoCrAlY. All the evaluated coatings performed better in this environment than the reference P91 and H800 steels.

## 1. Introduction

The durability and reliability of power generation equipment are important parameters affecting the stability of the energy market. In the case of components of combustion plants, such as boiler firewalls, heat exchangers, or pressure vessels of coal-fired power plants, the service life is often limited by the corrosion attack of an aggressive environment, where hot corrosion is especially a major issue determining the service life of said components [1,2,3,4,5,6,7,8,9,10,11,12,13,14]. Hot corrosion has been identified as a serious problem for many high-temperature aggressive environment applications, such as boilers [1,2,4,5,6,7,8,11,12,14], internal combustion engines, gas turbines [15,16], fluidized bed combustion boilers [17], industrial waste incinerators [18,19], etc.

The most dominant substance known to promote hot corrosion processes is Na_2_SO_4_, mainly owing to its very good temperature stability in a wide range of oxygen partial pressures [7]. Na_2_SO_4_ originates in oxidation reactions during the combustion of fuel containing both sulphur and sodium and possibly other alkali metals, such as potassium, which also form another corrosively aggressive sulphate attributed to the hot corrosion process—K_2_SO_4_ [1,7,8,10,20]. Some other major aggressive substances known for their negative attribution to hot corrosion process are chlorides and vanadates [7,8]. A major role in the formation of highly corrosive sulphates is played by sulphur; papers [21,22] discuss some aspects of the role of sulphur and its contribution to corrosive processes in high-temperature environments.

Volatile alkalic sulphates (primarily Na_2_SO_4_) and chlorides (such as NaCl and KCl) in the gaseous state react readily with other elements of fly ash in a high-temperature oxidizing environment during the combustion process. These low-melting-point compounds condense and form deposits on the surface of boiler components. This leads to the oxidation, sulfidation, and chlorination of the material and to high-temperature corrosion. The corrosion rate increases with rising temperature [1,2,3,7,8].

Ideally, a passivation oxide layer is formed on the surface of the material, acting as a diffusion barrier, and preventing further oxidation. Therefore, further growth of the oxide layer would require elements of the metal or oxidizing agent to be transported through the layer by solid state diffusion. In practice, the oxidation layer is often porous and cracked, which enables easier transport. In addition, ion diffusion is much faster through sulphates than through oxides. The molten salts act as a transport medium for the oxidizing agent into the material and for molten metal ions out of the material. In addition, chemical reactions are faster in the liquid state compared to the solid state. As far as the salt layer is concerned, the formation of the liquid state defines type I hot corrosion. In type I salt-induced hot corrosion, the temperature is higher than the melting temperature of the salt and therefore corrosion processes are significantly accelerated. A chemical reaction then takes place, which initially attacks the protective oxide film and continuously reduces the chromium (Cr) content of the substrate materials. As a result of the reduction in chromium content, oxidation of the substrate increases rapidly and porous deposits are formed [5,7,8]. The reaction producing metal chlorides or sulphides near the oxide/metal consumes metallic elements, especially chromium. The metal elements are determined to build or repair the passivating oxide layer. In an oxidizing atmosphere, the oxide scale protecting the base iron-based high-temperature resistant alloy is usually mostly composed of chromium oxides and iron oxides. If alloy contains aluminum, alumina (Al_2_O_3_) is also found in the oxide layer [8]. Solid salt deposits are deposited on the oxide scale and when the temperature rises above the melting point of the salt deposits, the salt mixture becomes liquid or partially liquid [1,2,3,4,5,6,7]. This is when the temperature threshold at hot corrosion type I begins to occur. The actual melting temperature depends on the salt mixture composition, but the melting temperature of the salt mixture is usually below melting temperature of its component due to eutectic character of the mixture [4,5,6,7,8,9]. 

During the initial stage of the corrosion process, when the salt deposit is melted, a molten sulphate salt film is formed on the oxide layer of the base material and the dissolution of oxides occurs (Equation (1) [2] for Cr_2_O_3_) and is promoted by an oxidizing atmosphere.
Cr_2_O_3_ (s) +3/2 O_2_ (g) +2 SO_4_^2−^ (l) = CrO_4_^2−^ (s) + 2 SO_3_ (g)(1)

Sodium chloride reacts with the oxides, releasing chlorine, which reacts further with the oxides to form volatile chlorides (Equation (2) shows a representative reaction with chromia, but basically the same reaction is for all presented metal oxides). [2]
8 NaCl (l) + 2 Cr_2_O_3_ (s) +5 O_2_ (g) = 4 Na_2_CrO_4_ (s) + 4 Cl_2_ (g)(2)

Chlorine in the gaseous state, Cl_2_, can rapidly penetrate through the oxide layer along cracks and reacts with material elements, such as chromium, forming volatile chlorides (Equation (3)) [3].
Cr (s) + 3/2 Cl_2_ (g) = CrCl_3_ (s)(3)

In the case of the thermal spray coating layer, volatile chlorides tend to diffuse along the splat boundaries to the interface of the coating and the substrate where they can oxidize again (Equation (4)). This restores chlorine and repeats the corrosion reaction. Numerous pits at the splat boundaries formed by volatile chlorides create another path for corrosive elements and facilitate the penetration of the corrosive environment into the material [2,7,8].
2 CrCl_3_ (g) + 3/2 O_2_ (g) = Cr_2_O_3_ (s) +3 Cl_2_ (g)(4)

Other material elements such as nickel or titanium are also subject to these reactions [1,2,3,4,5,7,8].

If the oxide layer on the surface of the material does not have protective properties, chlorides can substantially accelerate oxidation. Dispersed gaseous chlorine can penetrate the corrosion layer and the coating surface and can cause a high number of cracks and pores to form (the exemplar reaction with the iron in the base material is displayed in Equation (5)). The reaction of ferrous chloride with oxygen (Equation (6)) produces ferric oxide, which reacts again with sodium chloride (Equation (7)). A similar process also takes place for chromia (Equation (9)) [2,6].
Fe + Cl_2_ = FeCl_2_(5)
2 FeCl_2_ + 3/2 O_2_ = 2 Fe_2_O_3_ + Cl_2_(6)

Moreover, if there is no protective oxide layer on the surface, or when the protective layer has been damaged, such as in the case described above, molten sulphates readily penetrate towards the base material. The reaction of the sulphates with the metal oxides leads to the formation of low-melting-point alkali metal oxides (Equations (7) and (8)).
½ Fe_2_O_3_ + 3/2 NaSO_4_ = Fe_2_(SO_4_)_3_ + 3/2 Na_2_O(7)
½ Cr_2_O_3_ + 3/2 Na_2_SO_4_ = Fe_2_(SO_4_)_3_ +3/2 Na_2_O(8)
Fe_2_O_3_ + 3 SO_3_ = Fe_2_(SO_4_)_3_(9)

In the case of the corrosive environment containing only sulphates, sulphur oxide reacts with ferric oxide and forms ferric sulphate (Equation (9)). However, if the corrosive environment also contains chlorides, sulphur dioxide may also react with sodium oxide to form sodium sulphate and chlorine. Sulphates cause corrosion at splat boundaries compared to chlorides that cause internal corrosion. The presence of sulphur in the form of sulphates can cause internal alloy sulfidation under the protective oxide layer. Only a small amount of sulphides is formed in the corrosion layer [1,3,4,7,8].
SO_3_ +2^e−^ = SO_2_ + O^2−^(10)
NaCl + SO_2_ + O_2_ = Na_2_SO_4_ + Cl(11)

To prolong the service life of components, a surface treatment can be applied to protect the steel parts from corrosion in an aggressive environment. Among other deposition technologies, the thermal spray technology offers a feasible and economically acceptable way to coat the surface with a thick layer of corrosion resistant materials such as MCrAlY and MCrAlYX-type alloys. In MCrAlY(X) alloys, M stands either for Ni, Co, or both, where X is a minor alloying element, such as Re, Ta, Si, Hf, or often their combination. In recent decades, many scientific papers have been written on the topic of MCrAlY(X) hot corrosion resistance in many applications, e.g., coal-fired boilers [7,8], waste-to-energy boilers [18,19,22], biomass boilers [1,11,20], gas turbines [15,16], etc. In the experiment presented in this paper, several commercial MCrAlY(X) coatings deposited by HVOF technology are compared. In addition to these coatings, two conventional base materials that are commonly used for heat exchange tubes and firewalls of power plants are also tested—steel P91 and alloy 800 H. The aim of this study is to present a specific comparison of the potential benefits of increased corrosion resistance when using a coating in critically corroded areas of combustion plant heat exchanger surfaces compared to the corrosion resistance of the aforementioned conventional base materials of said components. The corrosive environment for the test is a salt mixture of 75% Na_2_SO_4_ and 25% NaCl, which is a simplified corrosive environment typical for the combustion chamber and flue gas path in energy producing boilers [1,6,7,8,11,19,21,22,23]. 

## 2. Materials and Methods

Four commercially available powders were used to coat the specimens. These powders were Amperit 410.001 (NiCoCrAlY) with particle size suitable for HVOF (−45 + 22) mm, Amperit 421.001 (NiCoCrAlTaReY) particle size (−45 + 22) mm, Amperit 405.001 (NiCoCrAlYHfSi) particle size (−45 + 22) mm, and Amperit 469.001 (CoCrAlYTaCSi) particle size (−53 + 20) mm. All coatings were deposited using HP/HVOF (High Pressure/High Velocity Oxygen Fuel) technology with JP-5000 torch from the TAFA Incorporated. Previously optimized spraying parameters were used for the coating application. Stainless high-temperature-resistant steel 1.4923 was used as a base material for coatings. Further, uncoated P91 steel and alloy 800 H samples were used as a reference specimen representing an uncoated firewall or heat exchange pipe in the same environment.

The substrate surfaces were degreased and grit-blasted prior to spraying to achieve proper adhesion between the coatings and the base materials. Brown corundum F22 with grain size (0.8 to 1.0 mm) was used as abrasive medium. The applied coatings thickness ranged from 250 to 320 μm.

Hot corrosion test was based on exposure to a corrosive salt mixture at 800 °C. The test procedure was as follows: the specimens were first polished to the surface roughness of Ra_max_. = 1 µm. The next step included cleaning with alcohol and heating in an oven at 250 °C. Heating is necessary for proper adhesion of the salt layer. For the simulation of hot corrosion conditions and result comparison, a mixture of salts of 75 wt.% Na_2_SO_4_ and 25 wt.% NaCl was chosen, which is approximately 55 mol% of Na_2_SO_4_ and 45 mol% of NaCl. This composition is close to the eutectic composition of NaCl/Na_2_SO_4_ that is about a 59:41 molar ratio. The melting temperature of NaCl/Na_2_SO_4_ eutectic mixture is 626 °C [9], and the mixture used in test presented in this paper is close enough to the eutectic point of this mixture to become sufficiently melted when exposed to 800 °C. Thus, the test was performed in more aggressive type I salt-induced hot corrosion [7,8,9]. This mixture of salts was mixed with alcohol and applied on the surface of evaluated specimens in the amount of (3–5) g/cm^2^. This step was followed by drying of the applied mixture for 3 h in an oven at 100 °C. Before testing in furnace, each specimen was weighed and the weight was subsequently measured after each test cycle. To compare the results easily, the number of cycles was set at 50. Each cycle consisted of one hour in silicon carbide furnace and subsequent cooling for 20 min at room temperature.

Metallographic evaluation was conducted using a scanning electron microscope EVO MA 15 (Carl Zeiss SMT, Oberkochen, Germany) equipped with Quantax EDS system XFlash^®^ 5010 (Bruker, Billerica, MA, USA). Exposed specimens were further evaluated by EDS elemental analysis to specify the changes in the chemical composition. It is important to mention that the specimen preparation for metallographic evaluation is not a standard procedure usually used after the corrosion testing, but it is an essential part of the microstructure study of thermal sprayed coatings. Corroded thermal sprayed coatings require special care when separating apart from the original specimen. Pure alcohol was used as a cutting coolant. The separated part was further rinsed again with alcohol. Lateral specimen sides and beveled sharp edges of coating were ground. It was also necessary to grind the bottom edge of the base material and remove any impurities to prevent the possible oxidation. Each specimen was further rinsed with alcohol and dried in warm air. The sample prepared in this way was finally cold-mounted in epoxy and polished. The corroded samples prepared in this manner were subsequently subjected to the analyses described above.

The coatings’ phase compositions were evaluated using X-ray diffraction (XRD), the D8 Discover powder diffractometer, Bruker, in Bragg–Brentano geometry with a 1D detector and CoKα radiation. The scanned region was from 15 to 100° 2θ with a 0.03° 2θ step size and with a 96 s counting time per step. The obtained diffraction patterns were subjected to quantitative Rietveld analysis performed in the TOPAS 5, which uses the so-called fundamental parameters approach. Crystalline phases in the XRD patterns were identified using the ICSD database.

## 3. Results and Discussion

### 3.1. Corrosion Kinetics

The high-temperature corrosion tests were evaluated using the thermogravimetric method of corrosion products kinetics. The individual specimens were weighed together with a ceramic cup after each testing cycle and weight change was recorded. Figure 1 shows the graph of cumulative weight gains per specified area unit of exposed coating. 

Figure 2 shows a parabolic law of weight gains in dependence on the number of cycles. Rapid weight gains were observed for all evaluated specimens during the first five cycles. This weight change was caused by the formation of the oxide layer and by the stabilization of the entire process/test. Further weight gains observed during the high-temperature corrosion test were caused by the formation of corrosion products. It can be consequently assumed that a specimen with the lowest weight gain showed the best resistance to the high-temperature corrosion. 

The thermogravimetric method of corrosion products kinetics proved that the CoCrAlYTaCSi coating provides the best corrosion protection of the tested coating materials. According to the results obtained, the NiCoCrAlTaReY coating and the NiCoCrAlY coating also provide some level of corrosion protection with approximately half the weight gains compared to the uncoated P91 steel reference specimen. The NiCoCrAlYHfSi coating showed the highest weight gains, indicating lower corrosion resistance in this environment, comparable to the uncoated P91 steel reference specimen. The uncoated P91 chromium-molybdenum steel (reference base material) showed approximately 2 to 3 times higher weight gains than NiCoCrAlY and NiCoCrAlTaReY resistance to high-temperature corrosion compared to the evaluated coatings. When comparing the two reference specimens, 800 H alloy without any surface protection showed significantly higher weight gain compared to the P91 steel specimen. The hot corrosion test of 800 H alloy showed significant weight gains, indicating severe corrosion attack. Figure 3 shows macroscopic cross-section images of all evaluated specimens after exposure to the corrosive environment of chloride salts. 

### 3.2. SEM/EDS Cross-Section Analysis

Figure 4 presents a macroscopic cross-section image of the NiCoCrAlYHfSi-coated specimen after exposure to the corrosive environment showing corrosion attack on all uncoated surfaces. Coating delamination is indicated by yellow dashed lines and cracks are indicated by blue arrows. The corrosion attack of uncoated sides further expanded through the substrate below the coating (green arrows) causing a partial loss of adhesion between the coating and the base material (Figure 5a). The resulting cavity was filled with corrosion products; see SEM images in Figure 5a,b. The SEM images in Figure 5d–f show a visible corrosion attack on the interface between the splats and in the individual splats as well. The corrosive environment has penetrated through the coating due to the porosity and attacked the base material. The formation of the corrosion products resulted in the coating deformation and the development of vertical cracks caused by stresses in the material (Figure 5b).

The principle of corrosion attack on the NiCoCrAlTaReY-coated specimen was similar to that for the NiCoCrAlYHfSi coating. The corrosion attack spread from the uncoated specimen sides under the entire coating and caused the coating delamination. The macroscopic cross-section image of the coating after the exposure to the corrosive environment in Figure 6 shows the corrosion attack on all uncoated specimen sides. The corrosion products on the specimen underside indicate that the base material was significantly attacked by corrosion in this area. The upper third of the coating shows initial corrosion attack of individual splats and their interfaces, as in Figure 7.

The loss of flatness and layering of the corrosion products along the specimen perimeter show that all uncoated surfaces of the NiCoCrAlY-coated specimen were corroded during exposure to the corrosive environment of chloride salts. The macroscopic image in Figure 8 shows a severe corrosion attack on the uncoated underside of the base material.

As with the NiCoCrAlYHfSi coating, this specimen also shows the corrosion attack on the uncoated side edges, which further spread through the base material under the coating. The corrosion attack caused the loss of adhesion between the coating and the base material and subsequent coating delamination, see the SEM images in Figure 9. The corrosion of the NiCoCrAlY coating attacked the splat interfaces and the splats themselves. The corrosion attack on the coating is shown in the SEM images in Figure 9d,e.

Since the CoCrAlYTaCSi coating showed the best resistance to the high-temperature corrosion within the evaluation by the thermogravimetric method, a more detailed analysis was performed. The macroscopic image of the CoCrAlYTaCSi coating in Figure 10a clearly shows the lowest corrosion attack in comparison with three others evaluated MCrAlY coatings. As previously described for the other coatings, the lateral corrosion attack of the base material spread under the coating.

The corrosion attack in the coating cross-section is shown in detail in the SEM analysis images in Figure 11. The metallographic cut of the coating cross-section after the exposure to the corrosive environment was divided into nine parts (see the macroscopic image in Figure 10a), and the entire coating cross-section was analyzed in detail. Figure 11 shows the SEM analysis images of the specimen left side (positions 1–4) and the specimen right side (positions 6–9). The specimen shows a strong corrosion attack spreading below the coating from both sides up to a distance of several millimeters, see Figure 11a–d,f–h. The peripheral coating areas show a chromium-depleted region across the entire coating width. Chromium has accumulated at a high density in corrosion products (the EDS image in Figure 12). Figure 12 presents a thin alumina layer visible in the corrosion products. The oxides of chromium and aluminum were detected by the EDS analysis of oxygen. In addition, iron originating from the base material was detected on the coating surface in the corrosion products. A strong presence of chlorine, sodium, and sulfur was reported at the coating-substrate interface in the delaminated areas (see the EDS image in Figure 12). These elements confirm the presence of the corrosive medium at the coating–substrate interface. The coating shows almost no attack in the central part, see Figure 10e. A thin corrosion layer was formed on the coating surface and occasionally attacked the coating leading to a small thickness.

According to EDS analysis of the central coating part, a thin oxide layer based on aluminum was formed on the surface and prevented from or slowed further corrosion attack, see Figure 13; according to the aluminum and oxygen maps shown in Figure 13, the oxide layer consists of alumina, which is to be expected for corrosion-resistant material in an oxidizing environment. It is the formation of a compact alumina layer that is desirable for the successful retardation of corrosion processes [7,24,25,26]. A structural change (particularly in splats interface) was observed only by a few splats below the coating surface. A deposit containing iron oxides (iron is not a part of the coating composition) was observed on the coating surface. It probably originates from the deposit of corrosion products from the base material.

The corrosion attack on the specimen from all sides can be derived from the layering of corrosion products around the whole specimen. Detailed images of the corrosion attack from the SEM analysis are shown in Figure 14. The uncoated P91 chromium-molybdenum steel specimen showed very low corrosion resistance during the high-temperature corrosion test; see the macroscopic image of the coating cross-section in Figure 15.

The 800 H alloy specimen showed a very low corrosion resistance in the high-temperature corrosion test in the selected environment. Figure 16 shows the macroscopic cross-section image of the specimen after exposure to the corrosive environment. The specimen lost its original rectangular shape due to the severe corrosion attack. The corrosion products were layered mainly on the coating upper part. Loose deposits of the corrosion products from the other sides of the specimen fell off during handling. Figure 17 shows SEM analysis of the cross-section images of corrosion attack on 800 H alloy.

### 3.3. XRD Analysis

A phase composition evaluation by XRD analysis of selected coatings after the corrosion test and steel P91 for reference is depicted in the Figure 18. The detected phases for each specimen are summarized in the Table 1. The analysis was carried out on the surface of the specimen and the penetration depth of the radiation used was approximately 10–30 μm. All the samples analyzed contained iron oxide phases in the form of Fe_2_O_3_ and/or Fe_3_O_4_, which originated from the P91 steel substrate material. In most cases, sodium sulphate Na_2_SO_4_ and sodium chloride NaCl, as remnants of the corrosive environment, were present in minority in all samples. Aluminum oxide Al_2_O_3_ was detected in the NiCoCrAlYHfSi, NiCoCrAlY, and CoCrAlYTaCSi coatings, which probably formed an oxide protective barrier. In addition, cobalt oxide in the form of Co_3_O_4_ was present in the CoCrAlYTaCSi coating.

### 3.4. Discussion

Based on the thermogravimetric analysis method of corrosion kinetics, the CoCrAlYTaCSi coating applied by technology can be identified as the coating that best resisted corrosion in a given environment. The worst corrosion resistance of tested coatings was the NiCoCrAlYHfSi coating, for which higher mass gains were observed throughout the test. The uncoated chromium-molybdenum steel P91 had comparable corrosion behavior to the NiCoCrAlYHfSi-coated specimen. Significant mass increments were observed for the uncoated 800 H alloy, which is widely used in practice in this corrosive environment, indicating severe corrosion attack.

From the evaluation of the microstructure in the specimen cross-section by OM and SEM, according to the loss of flatness and the appearance of corrosion products, it is clear that corrosion attack occurred on all uncoated surfaces in all specimens. The corrosion mechanism was very similar for the NiCoCrAlYHfSi, NiCoCrAlY, NiCoCrAlTaReY, and CoCrAlYTaCSi coatings. Lateral attack propagated through the substrate material below the spray–substrate interface towards the center from the spray edges. Consequently, partial or total delamination of the coating was observed. The coating surface itself was hardly affected. The interface between the splats and the individual splats were attacked to some extent in all coatings. Due to the open porosity in the NiCoCrAlYHfSi coating, the corrosive environment penetrated through the coating to the substrate material. The subsequent formation of corrosion products caused deformation of the coating and the development of vertical cracks.

OM and SEM analyses further show that the CoCrAlYTaCSi coating had the least corrosion attack compared to the other tested samples. The specimen shows strong attack spreading under the coating from both sides up to a distance of several millimeters. Above the attacked part of the substrate material there is a newly formed coating structure with attack at the grain boundary and the splats themselves. At the edges of the coating there is an area of depleted chromium throughout the width of the coating, which was concentrated in high density in the area of the corrosion plumes, where a thin layer of aluminum was also visible. According to the EDS analysis of oxygen, these are their oxides. EDS analysis revealed the presence of corrosion medium elements in the coating volume and at the coating–substrate interface in the area of coating delamination. There was an increased presence of alumina at the interlaminate interface in the left edge portion of the coating. In addition, chromium depletion occurred at the interlaminar interface. In the central part, the coating was almost free of attack. However, the optical microscope images showed the corrosion attack on the substrate material began in the central part of the specimen. A thin aluminum and cobalt-based oxide layer had formed on the surface of the coating in the central part, which probably prevented further attack. A deposit containing iron oxides (which is not present in the coating) was observed across the width of the sample in the area of corrosion fumes on the surface of the coating, and is therefore probably a deposit of corrosion products from the underlying material [7,11,23,25].

The uncoated P91 steel showed significant attack during the high-temperature corrosion test, which was accompanied by a layering of corrosion products around the entire specimen. The corrosion attack was even more pronounced in the 800 H alloy specimen. The layering of corrosion fumes occurred mainly in the upper part of the coating and the specimen also lost its original rectangular shape.

The results of the phase composition evaluation revealed the presence of iron oxide in the form of Fe_2_O_3_ and/or Fe_3_O_4_ in all samples analyzed, which originated from the P91 steel base material. In most cases, sodium sulphate Na_2_SO_4_ and sodium chloride NaCl, as remnants of the corrosive environment, were present in the minority in all samples. Aluminum oxide Al_2_O_3_ was detected in the NiCoCrAlYHfSi, NiCoCrAlY, and CoCrAlYTaCSi coatings, which probably formed an oxide protective barrier. In addition, cobalt oxide in the form of Co_3_O_4_ was present in the CoCrAlYTaCSi coating. The XRD analysis of the phase composition shows that the coatings themselves hardly reacted with the corrosive environment [7,19,21,23,25]. 

## 4. Conclusions

The aim of this study was to evaluate the resistance of selected thermally sprayed coatings against the high-temperature corrosion in the aggressive environment of 75% Na_2_SO_4_ and 25% NaCl salt mixture and demonstrate a comparison of the potential increase in corrosion resistance when applying these selected commercial coatings. For this reason, in addition to the four commercial MCrAlY(X) coatings, P91 steel and 800 H alloy (both in the form of bulk material) are also tested in the present corrosion test as examples of some conventional base materials used in power equipment. 

The results of thermogravimetric evaluation of corrosion products kinetics, supported by the SEM and EDS observation of tested specimens, show that the corrosive environment of chloride salts is very aggressive for many common materials. The CoCrAlYTaCSi-coated specimen exhibited the best corrosion resistance to this environment. The tested sample showed a relatively integral protective layer of alumina after fifty 1-h cycles at 800 °C, which seems to have fulfilled its protective function against penetration of the aggressive environment of alkali metal sulphates and chlorides. Based on the test performed, the CoCrAlYTaCSi coating can be recommended as a potentially suitable surface protection for power plant boiler environments dominated by alkaline sulphates and chlorides. 

All specimens experienced corrosion attack on all uncoated surfaces and the coated specimens exhibited coating delamination to some extent. The observed corrosion mechanisms of the NiCoCrAlYHfSi, NiCoCrAlY, NiCoCrAlTaReY, and CoCrAlYTaCSi coatings were found to be similar. Lateral corrosion attack spread through the base material to the coating–substrate interface from the edges towards the center of the samples. The corrosion attack on splats and their interfaces was observed on all coatings. Uncoated P91 steel exhibited decent test results roughly comparable to the NiCoCrAlYHfSi-coated specimen (worst resistance among tested coatings), but since potential industrial utilization of HVOF-sprayed NiCoCrAlYHfSi coating on P91 would not probably lead to any noticeable increase in corrosion resistance and would only serve to add some minor thickness to coated firewall or tube, it cannot be recommended for a corrosion environment with the dominant influence of sodium sulphates and chlorides. The corrosion attack on the 800 H alloy specimen proved that this material is completely unsuitable for applications in aggressive environments with chloride salts.

## Figures and Tables

**Figure 1 materials-16-04492-f001:**
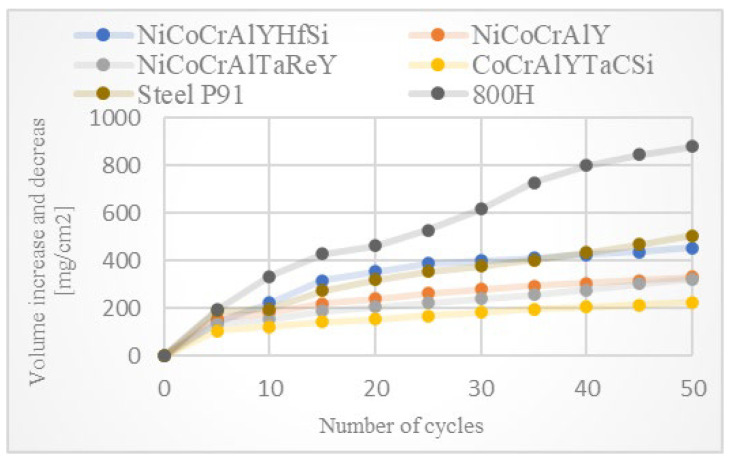
The graph of weight gains per area unit related to the number of cycles for all evaluated coatings subjected to 50 cycles of the high-temperature corrosion test in 75% Na_2_SO_4_ and 25% NaCl at 800 °C. Reference to uncoated boiler firewall or heat exchange tube represents bulk steel P91 and alloy 800 H (brown and dark grey plots respectively).

**Figure 2 materials-16-04492-f002:**
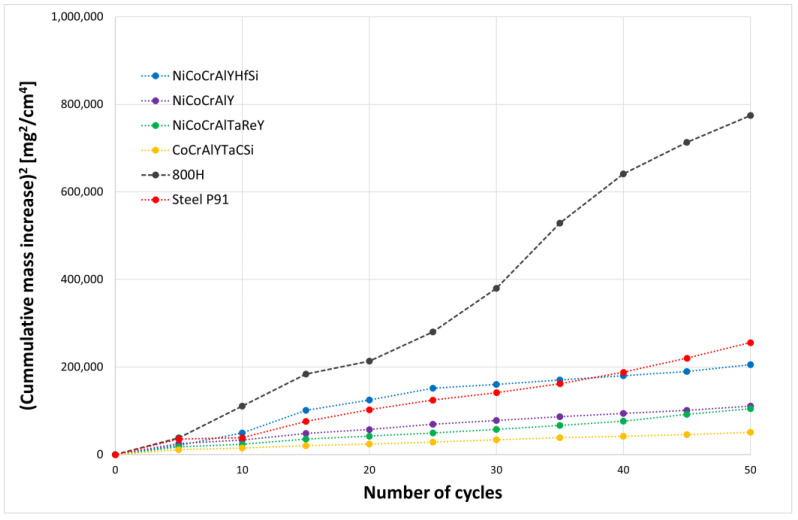
The parabolic graph of weight gains per area unit related to the number of cycles for all evaluated coatings exposed to 50 cycles of the high temperature corrosion test in 75% Na_2_SO_4_ and 25% NaCl at 800 °C. References are uncoated boiler firewall or heat exchange tube representing bulk steel P91 and alloy 800 H (brown and dark grey plots, respectively).

**Figure 3 materials-16-04492-f003:**
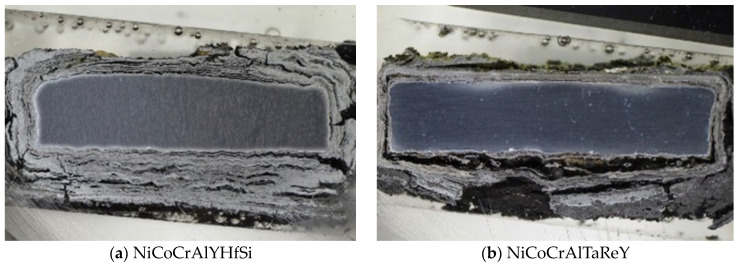
The macroscopic images of the coatings (**a**) NiCoCrAlYHfSi, (**b**) NiCoCrAlTaReY, (**c**) NiCoCrAlY, (**d**) CoCrAlYTaCSi, (**e**) reference steel P91 and (**f**) alloy 800 H showing cross-sections after the high-temperature corrosion test.

**Figure 4 materials-16-04492-f004:**
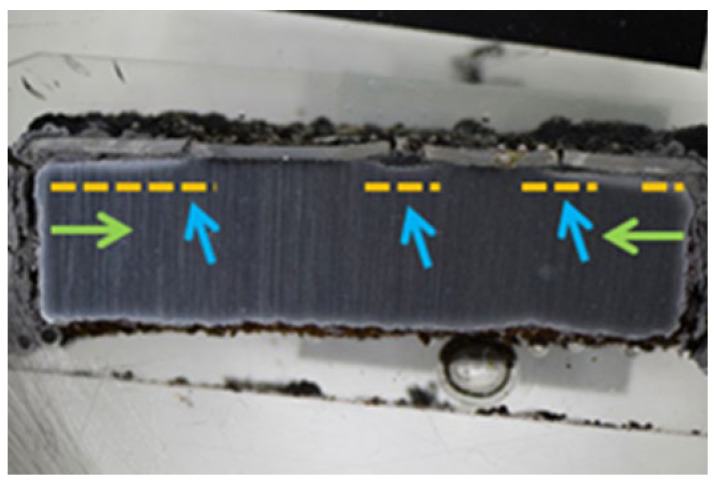
Macroscopic cross-section image of the NiCoCrAlYHfSi-coated specimen after exposure to the corrosive environment.

**Figure 5 materials-16-04492-f005:**
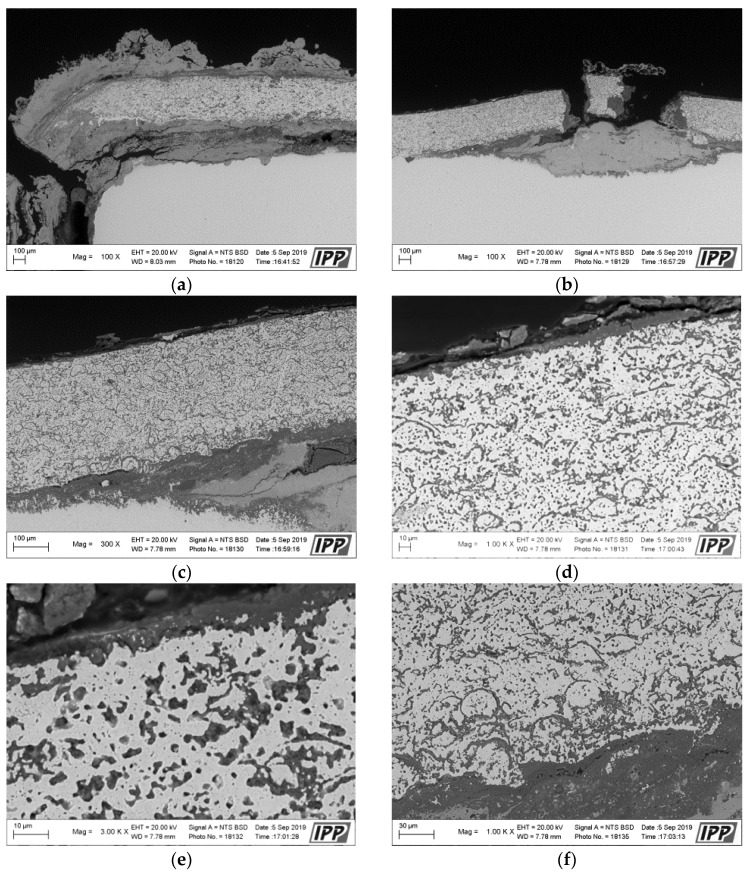
SEM analysis images in the cross-section of the NiCoCrAlYHfSi-coated specimen after the exposure to the corrosive environment: (**a**) left side area with 100× magnification, (**b**) center area with 100× magnification, (**c**) center area with 300× magnification, (**d**) center area—coating surface with 1000× magnification, (**e**) center area—coating surface with 3000× magnification, (**f**) center area—coating underside with 3000×.

**Figure 6 materials-16-04492-f006:**
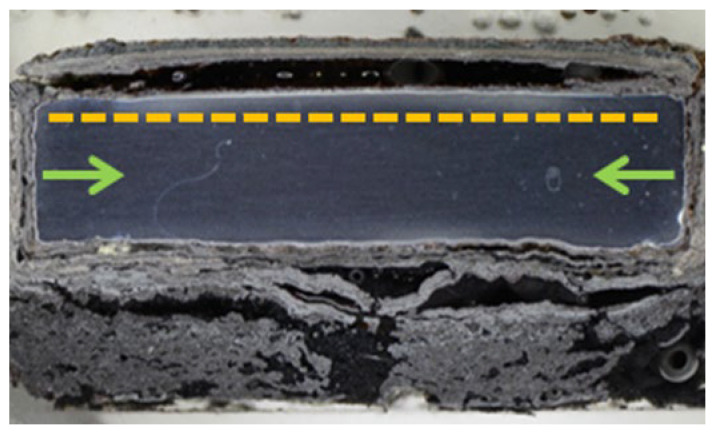
Macroscopic cross-section image of the NiCoCrAlTaReY-coated specimen after exposure to the corrosive environment.

**Figure 7 materials-16-04492-f007:**
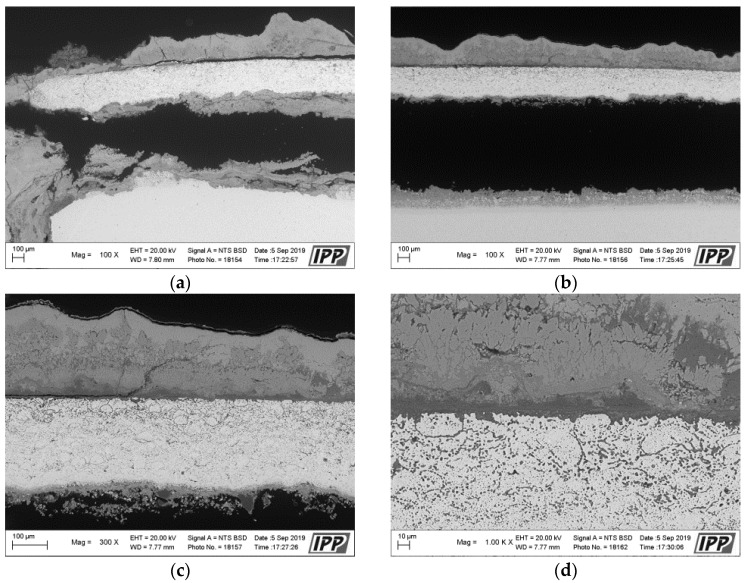
SEM analysis images in the cross-section of the NiCoCrAlTaReY-coated specimen after the exposure to the corrosive environment: (**a**) left side area with 100×, (**b**) center area with 100×, (**c**) center area with 300×, (**d**) center area—coating surface with 1000×, (**e**) center area—coating surface with 3000×, (**f**) center area—coating underside with 3000×.

**Figure 8 materials-16-04492-f008:**
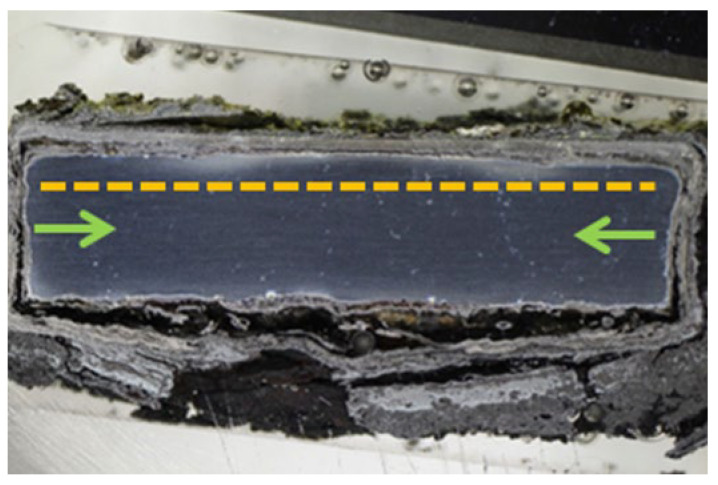
Macroscopic cross-section image of the NiCoCrAlY-coated specimen after the exposure to the corrosive environment.

**Figure 9 materials-16-04492-f009:**
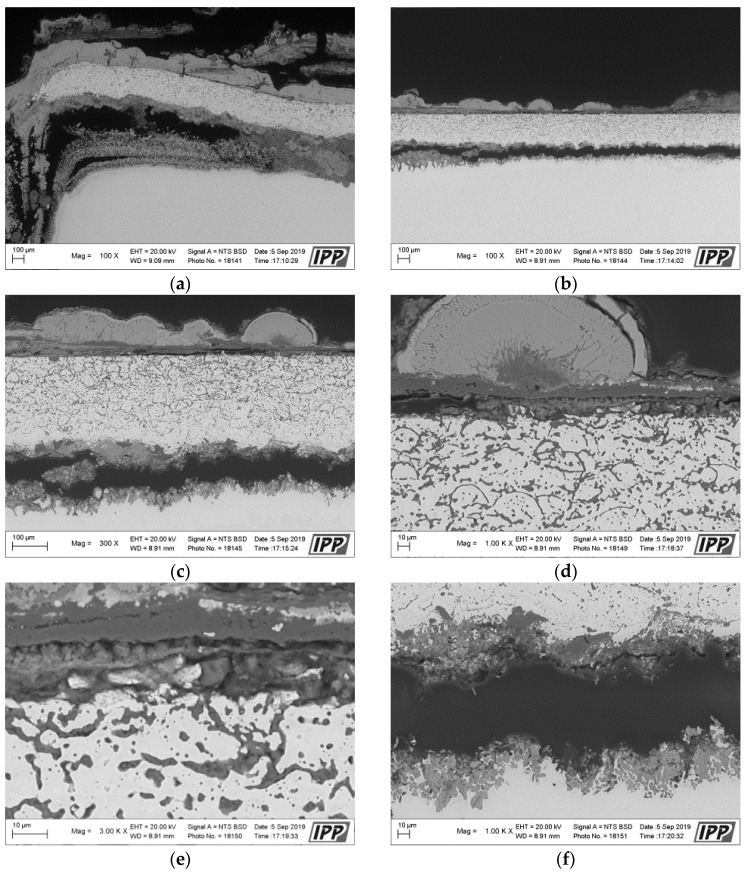
SEM analysis images in the cross-section of the NiCoCrAlY-coated specimen after exposure to the corrosive environment: (**a**) left side area with 100×, (**b**) center area with 100×, (**c**) center area with 300×, (**d**) center area—coating surface with 1000×, (**e**) center area—coating surface with 3000×, (**f**) center area—coating underside with 3000×.

**Figure 10 materials-16-04492-f010:**
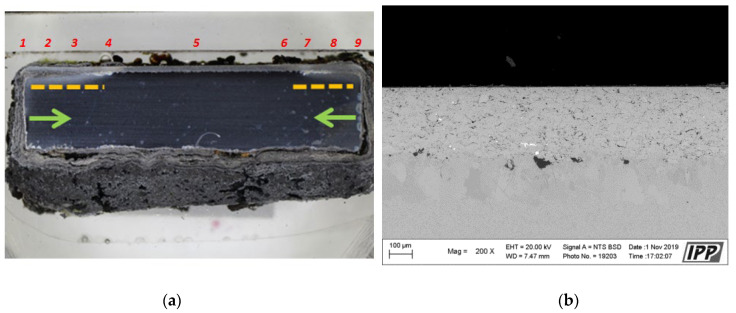
(**a**) Macroscopic cross-section image of the CoCrAlYTaCSi-coated specimen after the exposure to the corrosive environment, area 5—(**b**) SEM analysis image with 200×—the coating center taken from an area 5 (red number 5 on the image (**a**)).

**Figure 11 materials-16-04492-f011:**
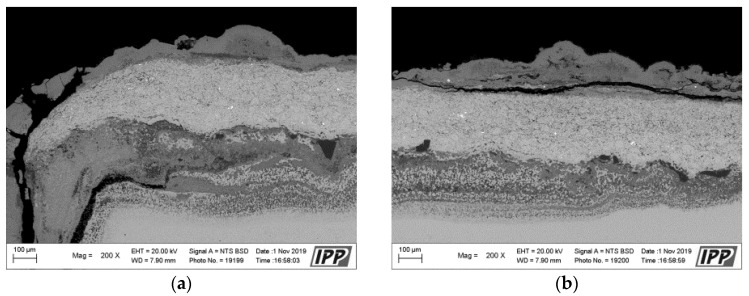
(**a**–**h**) SEM analysis images of the cross-section of the CoCrAlYTaCSi-coated specimen (coating with best results in the test), after the exposure to the corrosive environment, from left to right.

**Figure 12 materials-16-04492-f012:**
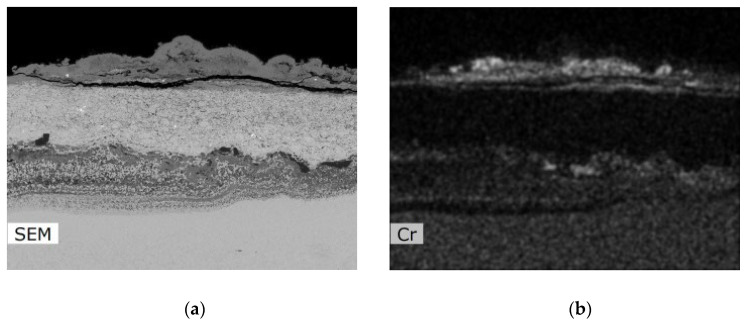
EDS analysis images of the CoCrAlYTaCSi-coated specimen (coating with best results in the test); (**a**) SEM image of the area of EDS analyzes, (**b**) EDS map of Cr content, (**c**) EDS map of Al content, (**d**) EDS map of Fe content, (**e**) EDS map of O content, (**f**) EDS map of Cl content, (**g**) EDS map of S content, (**h**) EDS map of Na content.

**Figure 13 materials-16-04492-f013:**
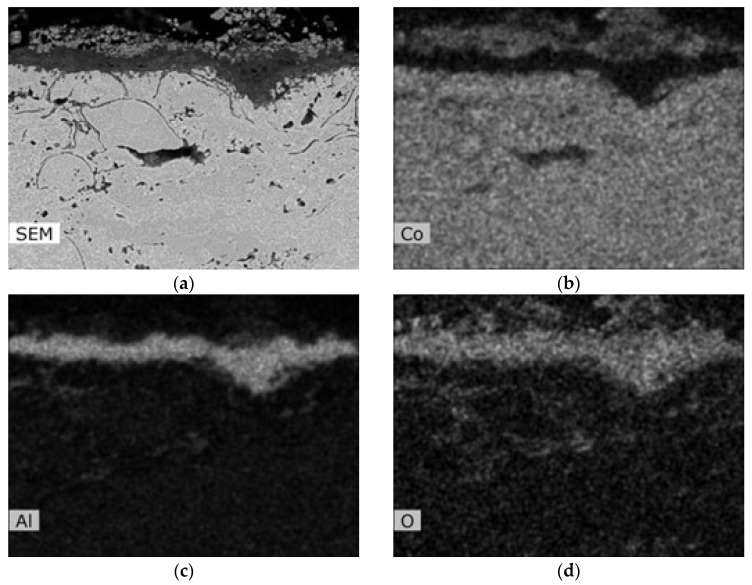
EDS analysis images of the CoCrAlYTaCSi-coated specimen (coating with best results in the test)—central part; (**a**) SEM image of the area of EDS analyzes, (**b**) EDS map of Co content, (**c**) EDS map of Al content and (**d**) of O content show presence of continuous protective aluminum-rich oxide layer on the surface.

**Figure 14 materials-16-04492-f014:**
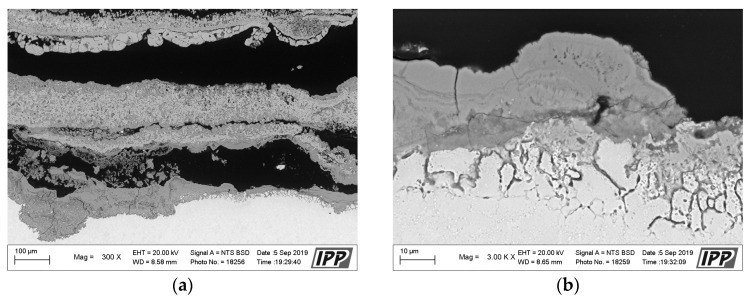
EDS analysis images of the P91 steel specimen after exposure to the corrosive environment at (**a**) 300×, (**b**) 3000×.

**Figure 15 materials-16-04492-f015:**
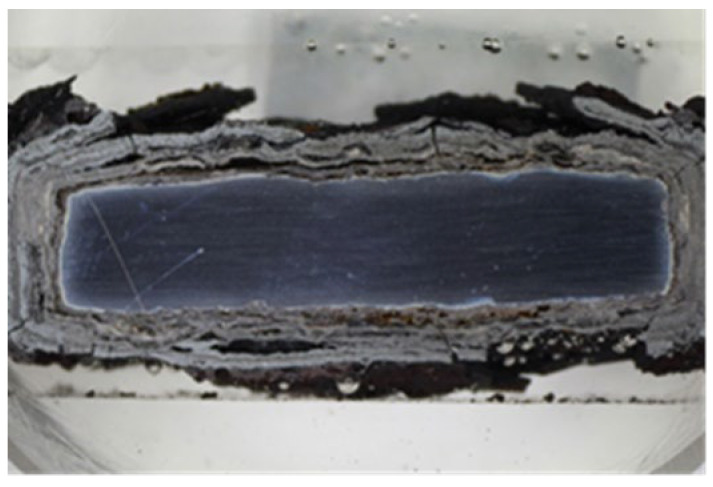
Macroscopic image of the cross-section of the P91 steel specimen after exposure to the corrosive environment.

**Figure 16 materials-16-04492-f016:**
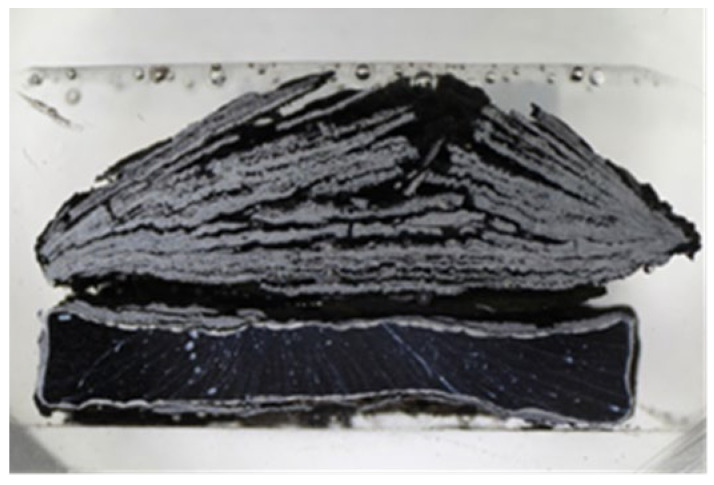
Macroscopic cross-section image of the 800 H alloy specimen after exposure to the corrosive environment.

**Figure 17 materials-16-04492-f017:**
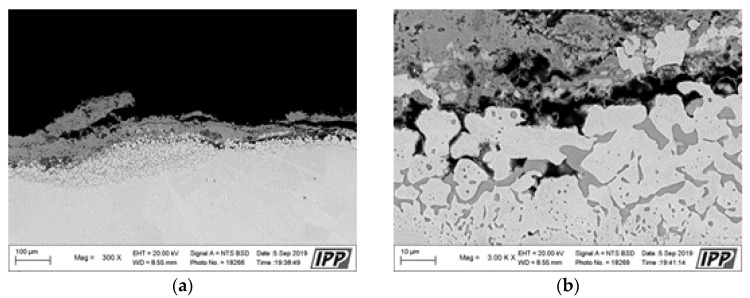
SEM analysis images of the cross-section of the 800 H alloy specimen after exposure to the corrosive environment with magnification of (**a**) 300×, (**b**) 3000×.

**Figure 18 materials-16-04492-f018:**
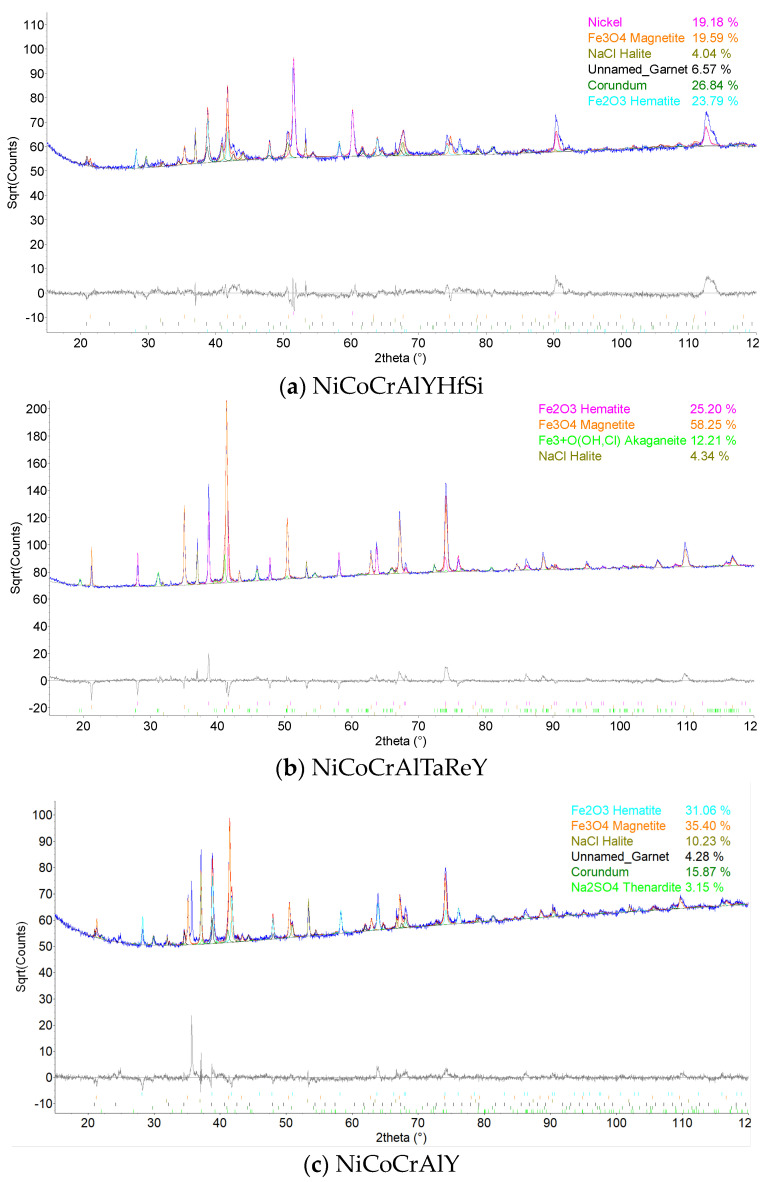
Phase composition analysis by XRD: (**a**) NiCoCrAlYHfSi, (**b**) NiCoCrAlTaReY, (**c**) NiCoCrAlY, (**d**) CoCrAlYTaCSi, (**e**) P91 steel.

**Table 1 materials-16-04492-t001:** Phase composition of specimen surface after exposure to the corrosion environment determined using XRD analyses.

Coating/Specimen	Major Phases	Minor Phases
NiCoCrAlYHfSi	Al_2_O_3_, Fe_2_O_3_, Fe_3_O_4,_ Ni	Garnet, NaCl
NiCoCrAlTaReY	Fe_3_O_4_, Fe_2_O_3_, Fe_3_ + O (OH, Cl)	NaCl
NiCoCrAlY	Fe_3_O_4_, Fe_2_O_3_, Al_2_O_3,_ NaCl	Garnet, Na_2_SO_4_
CoCrAlYTaCSi	Al_2_O_3_, hcp Co, Co_3_O_4_, NaCl	Fe_3_O_4_, fcc Co, Fe_2_O_3_
P91 steel	Fe_2_O_3_	Na_2_SO_4_, NaCl

## Data Availability

Data are contained within the article or are available on request from the corresponding author.

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
