# Peer review of "High-Temperature Corrosion Behavior of Selected HVOF-Sprayed Super-Alloy Based Coatings in Aggressive Environment at 800 °C"

_materials, 2023, doi:10.3390/ma16124492_

Round 1

Reviewer 1 Report

Manuscript provides very good concept.

  • Minor changes are still needed in the English language.

Author Response

Dear Reviewer,

we thank you for the positive report on our work.

We have done minor editing of English language. We will ensure additional English editing prior final publication.

Best regards
Marek Vostrak on behalf of the collective of authors

Author Response

Dear Reviewer,

we thank you for your review and for interesting suggestions and questions.

Here are our reactions to your questions:

  1. What is the novelty of the work?

The goal of the work and its novelty was commented in the manuscript.

The aim of this study is to present a specific comparison of the potential benefits of in-creased corrosion resistance when using a coating in critically corroded areas of combustion plant heat exchanger surfaces compared to the corrosion resistance of the aforementioned conventional base materials of said components.

  1. How the results obtained in the work relate to the results obtained by other authors?

We have revised and added some comments to other results to the results and discussion part and conclusion. All changes are highlighted in the revised manuscript.

  1. What was the thickness of coatings?

Thickness of the coating range from 250 to 320 μm. It is added to the Materials and Methods chapter.

  1. Have the coating adhesion tests been performed?

We have not performed the adhesion tests of the applied coating. We standardly do the adhesion test according to norm . By most our HVOF applied coating have very good adhesion exceeding the possibilities of the glue. As the adhesion of these HVOF sprayed coating is sufficient for intended application we have not included the adhesion test to the study.

  1. It seems that the obtained coatings slightly protect the substrates against the high temperature corrosion. How do the authors intend to improve their properties? What are the research plans for the future?

Currently we focus our research activities in this area on combined stress and wear analysis. As in the potential application it is rarely only corrosion attack of the coating, usually there is also erosion or other types of wear. We are conducting an experiment in which we combine erosion wear test for coating in different stages of corrosion cycle. We are currently also focused more on coating applied by mobile thermal spraying methods as TWAS.

Further we have updated the introduction part and added more topic-related references, further we have revived the results and discussion part and conclusion. All changes are highlighted in the revised manuscript.

We hope that we have improved the quality of our work. In case of further questions and suggestions we are open to further additions and modifications

Best regards
Marek Vostrak on behalf of the collective of authors

Reviewer 3 Report

The present paper studies the high-temperature corrosion resistance of four thermal spray coatings under severe conditions. The study is well-planned and interesting.

However, the manuscript has some issues that need to be improved.

(1) In the Introduction, the author outlines principles of high-temperature corrosion at length, but fails to highlight the paper's significance and originality, the author should elaborate the innovative points of this research.

(2) Since this study focuses on the corrosion behaviors of coatings, comparing the coated specimens with two uncoated materials P91 and H800 would make readers a little confused. If to prove the protection of these coatings, it’s more reasonable to use the uncoated base alloy as a reference specimen, rather than two other materials.

 (3) In sec 3.1, all analyses can be obtained based on Fig.1. The deeper and unique information reflected in Fig.2 is not described.

(4) There are many writing and formatting errors in this manuscript. For example, (a) the molecular formula of sodium sulfate should be written as Na2SO4 instead of Na2SO4; (b) Some paragraphs have two characters indented at the beginning, while other paragraphs do not; (c) There is a syntax error in the sentence “molten sulphates easily penetrate into the contact with…….”; (d) The figures and corresponding titles should be on the same page.

Please See comments 

Author Response

we thank you for your review and for interesting suggestions and questions.

Here are our reactions to your questions:

(1) In the Introduction, the author outlines principles of high-temperature corrosion at length, but fails to highlight the paper's significance and originality, the author should elaborate the innovative points of this research.

We have revised the introduction chapter of the manuscript, added more topic related references and also we have added the comments on the aim of the research. We have also revised the results and discussion part and conclusion. All changes are highlighted in the revised manuscript.

(2) Since this study focuses on the corrosion behaviors of coatings, comparing the coated specimens with two uncoated materials P91 and H800 would make readers a little confused. If to prove the protection of these coatings, it’s more reasonable to use the uncoated base alloy as a reference specimen, rather than two other materials.

We have selected these uncoated materials as a reference as they are sometimes used in energy industry for high temperature applications without coating protections and they should have some level of corrosion resistance. The reasoning behind choosing these materials was to compare behaviour of these often use but uncoated alloys with possible protection potential of selected coatings. If we would use the uncoated base alloy it would behave very badly in corrosion test.

 (3) In sec 3.1, all analyses can be obtained based on Fig.1. The deeper and unique information reflected in Fig.2 is not described.

We have added some new comments to the description of Corrosion kinetics to the manuscript.

(4) There are many writing and formatting errors in this manuscript. For example, (a) the molecular formula of sodium sulfate should be written as Na2SO4 instead of Na2SO4; (b) Some paragraphs have two characters indented at the beginning, while other paragraphs do not; (c) There is a syntax error in the sentence “molten sulphates easily penetrate into the contact with…….”; (d) The figures and corresponding titles should be on the same page.

We have done some revisions and editing of these errors in the manuscript as well as minor editing of English language. We will ensure additional editing once the manuscript will include all revisions based on review process prior final publication.

We hope that we have improved the quality of our work. In case of further questions and suggestions we are open to further additions and modifications.

Best regards
Marek Vostrak on behalf of the collective of authors

Round 2

Reviewer 2 Report

The paper in this form may be published..